Western scrub-jays do not appear to attend to functionality in Aesop’s Fable experiments

Logan Corina J. 1 5 cl417@cam.ac.uk
Harvey Brigit D. 2
Schlinger Barney A. 2 3 4
Rensel Michelle 2
1 SAGE Center for the Study of the Mind, University of California , Santa Barbara, CA , United States
2 Department of Integrative Biology and Physiology, University of California , Los Angeles, CA , United States
3 Laboratory of Neuroendocrinology, Brain Research Institute, University of California , Los Angeles, CA , United States
4 Department of Ecology and Evolutionary Biology, University of California , Los Angeles, CA , United States
5 Current affiliation: Department of Zoology, University of Cambridge , Cambridge , United Kingdom
Vonk Jennifer
Electronic publication date: 2016 Feb 23
Publication date: 2016
Volume: 4
Electronic Location ID: e1707
Received 2015 Dec 28; Accepted 2016 Jan 28
Copyright: ©2016 Logan et al.
Copyright year: 2016
Copyright holder: Logan et al.
License: This is an open access article distributed under the terms of the Creative Commons Attribution License, which permits unrestricted use, distribution, reproduction and adaptation in any medium and for any purpose provided that it is properly attributed. For attribution, the original author(s), title, publication source (PeerJ) and either DOI or URL of the article must be cited.
License URL: https://creativecommons.org/licenses/by/4.0/

Keywords: Western scrub-jay, Aesop’s Fable, Comparative cognition, Flexible behavior, Non-caching paradigm

Funding: National Institute of Mental Health MH061994 This research was funded by the SAGE Center for the Study of the Mind at the University of California Santa Barbara (CJL) and by an RO1 grant from the National Institute of Mental Health (grant number: MH061994; BAS). The funders had no role in study design, data collection or analysis, decision to publish, or preparation of the manuscript.

==============================
Western scrub-jays are known for their highly discriminatory and flexible behaviors in a caching (food storing) context. However, it is unknown whether their cognitive abilities are restricted to a caching context. To explore this question, we tested scrub-jays in a non-caching context using the Aesop’s Fable paradigm, where a partially filled tube of water contains a floating food reward and objects must be inserted to displace the water and bring the food within reach. We tested four birds, but only two learned to drop stones proficiently. Of these, one bird participated in 4/5 experiments and one in 2/5 experiments. Both birds passed one experiment, but without attending to the functional differences of the objects, and failed the other experiments. Scrub-jays were not motivated to participate in these experiments, suggesting that either this paradigm was ecologically irrelevant or perhaps their flexibility is restricted to a caching context.

Introduction

Western scrub-jays (Aphelocoma californica; hereafter referred to as scrub-jays) are known for their highly discriminatory and flexible behaviors in a caching (food storing) context. For example, scrub-jays prefer to recover perishable food items sooner than non-perishable items (Clayton, Yu & Dickinson, 2001), they plan what they want for breakfast the next morning (Raby et al., 2007), and scrub-jays with prior experience stealing other’s caches use cache protection strategies (Dally, Emery & Clayton, 2006, see review in Grodzinski & Clayton, 2010). However, it is unknown whether such abilities are restricted to a caching context—the context in which these abilities evolved (Grodzinski & Clayton, 2010).

To begin to answer this question, we tested scrub-jays in a non-caching context using the Aesop’s Fable paradigm. In this paradigm, clear tubes that are partially filled with water contain a floating food reward that can only be reached by inserting objects into the tube to raise the water. These experiments investigate how individuals solve problems by giving them objects or tubes that vary in their functionality (Bird & Emery, 2009a; Taylor et al., 2011; Cheke, Bird & Clayton, 2011; Jelbert et al., 2014; Logan et al., 2014; Jelbert, Taylor & Gray, 2015). If individuals choose the functional option significantly more than the non-functional option, it indicates that they might have a causal understanding of the properties of objects and substrates. Alternatively, they might have an innate bias toward the more functional object or they might associate the movement of the food in the tube (e.g., rising toward the top of the tube with every object dropped in) with the particular substrate or object type. To probe these explanations and understand how individuals solve these tasks, particularly whether they use causal cognition, tasks are often presented where the solution violates causal explanations and thus are counter-intuitive in this context. These tests involve the presentation of an apparatus with a hidden mechanism such that individuals must rely on arbitrary cues (e.g., color cues) to solve the problem because there are no causal cues to attend to. Therefore, if an individual uses causal cues when solving water tube tasks, they should fail the counter-intuitive experiments (Cheke, Bird & Clayton, 2011; Jelbert et al., 2014; Logan et al., 2014). Failure to learn to associate an arbitrary cue over the course of 20 trials in experiments that function in such a way as to violate causal expectations is interpreted as evidence that individuals rely to some degree on causal information to solve water tube tasks.

In all other corvid species tested (rooks, New Caledonian crows, and Eurasian jays) using the Aesop’s Fable paradigm, at least some individuals successfully solved the tasks (Table 1, see Jelbert, Taylor & Gray, 2015 for a full overview). These individuals were sensitive to the functional properties of objects and substrates because they preferred the more functional option to gain a food reward. Almost all individuals failed the counter-intuitive colored U-tube test, which suggests they might rely to some degree on causal cues to solve water tube tasks. The only non-corvid bird tested so far, the great-tailed grackle (Quiscalus mexicanus), was also successful in Aesop’s Fable experiments and two individuals changed their preferences when circumstances changed, indicating behavioral flexibility (Logan, 2015a). Of these species, the Eurasian jay is the only other caching specialist and it exhibited flexibility outside of a caching context (Brodin & Lundborg, 2003; Pravosudov & De Kort, 2006). This leaves an open question of whether scrub-jays can apply their flexibility outside of a caching context.

Table 1 Summary of results from previous Aesop’s Fable experiments.

A summary of the main results from previous tests on birds with varying degrees of caching specialization and tool using abilities. For a more detailed summary see Jelbert, Taylor & Gray (2015).

Bird species	Cacher?	Tool user?	n	n stone droppers	Water tube experiments	Citation	
Rook (Corvus frugilegus)	Non-specialista	Nod	4	4	4 used just enough stones to reach the food	3 preferred large rather than small stones	3 preferred water over sawdust tube					Bird & Emery (2009a)	
Eurasian jay (Garrulus glandarius)	Specialista	Noe	5	4	2 preferred baited over unbaited tube	2 preferred sinking over floating objects	2 preferred liquid over solid or empty tube	0 preferred connected apparatus when mechanism was hidden				Cheke, Bird & Clayton (2011)	
New Caledonian crow (Corvus moneduloides)	Likelyb	Yesb	5	4	4 used just enough stones to reach the food	The group preferred large rather than small stones	The group preferred liquid over solid or empty tube	The group preferred heavy over light objects				Taylor et al. (2011)	
“			6	6	5 preferred water over sand tube	6 preferred sinking over floating objects	5 preferred solid over hollow objects	0 preferred narrow over wide tube	4 preferred wide over narrow tube	0 preferred connected apparatus when mechanism was hidden		Jelbert et al. (2014)	
“			8	6	3 preferred water over sand tube	6 preferred sinking over floating objects	6 preferred solid over hollow objects	4 preferred narrow over wide tube	3 preferred wide over narrow tube	1 preferred connected apparatus when mechanism was hidden	0 preferred connected apparatus when mechanism was exposed	Logan et al. (2014)	
Great-tailed grackle (Quiscalus mexicanus)	Noc	Noc	8	6	4 preferred more functional heavy over less functional light objects	2 switched from preferring heavy to no preference when only light object was functional	0 preferred narrow over wide tube					Logan (2015a)	
Notes.

n the total number of birds that began stone dropping training

n stone droppers the total number of birds that passed stone dropping training

a Brodin & Lundborg (2003)

b Hunt (2000a), Hunt (2000b) and Kenward et al. (2006).

c Not reported in Skutch, (1954) (referred to as boat-tailed grackles).

d Bird & Emery, (2009b): not in the wild, but they can make and use tools in the lab.

e Not reported in Lefebvre, Nicolakakis & Boire (2002).

We gave scrub-jays five Aesop’s Fable experiments that have been conducted on other bird species to make their performance comparable. Although some species that have passed these experiments make and use tools in the wild, non-tool using birds are also able to pass these tests (Table 1). Therefore, these experiments should be within the capacity of the non-tool using scrub-jays (Lefebvre, Nicolakakis & Boire, 2002). In Experiment 1 (Water vs. Sand), one tube was partially filled with water and the other with sand; stones were available to solve the task by dropping them into the water tube (Bird & Emery, 2009a; Bird & Emery, 2009b; Taylor et al., 2011; Jelbert et al., 2014; Logan et al., 2014). Experiment 2 (Heavy vs. Light) consisted of one water tube with more functional heavy objects and less functional light objects, while Experiment 3 (Heavy vs. Light Magic) was the same except the heavy objects became non-functional because they stuck to a magnet while the light objects became the functional option because they fell past the magnet (Logan, 2015a). Behavioral flexibility, the ability to quickly change preferences when the task changes, would be demonstrated if individuals that preferred heavy objects or had no object preference in Experiment 2 changed their preference to either no preference or to preferring light objects in Experiment 3.

Experiment 4 (Colored U-tube) was counter-intuitive, consisting of two differently colored apparatuses, each with a small tube containing food, but too small to insert stones, and a large tube that could accommodate stones (Logan et al., 2014). One apparatus had a hidden connector tube under the lid that connected the large and small tubes such that if a stone was dropped in to the connected large tube, the water levels would rise in both the large and small tubes. To succeed, the bird must associate the color of the connected apparatus or the movement of the food with receiving a food reward. This experiment was modified from its previous version (in Logan et al. 2014): we made each apparatus more visually distinct through expanding the color cues and shapes to include the whole apparatus and both tubes. These changes should facilitate the perception that both tubes belonged to one apparatus rather than being separate. Experiment 5 (Uncovered U-tube) was the same as Experiment 4 except all color cues were removed and the connector tube exposed so the bird could see how the apparatus worked (Logan et al., 2014). Additionally, the water in the large tubes was tinted with food coloring such that when stones were dropped into the connected apparatus, water in the connected small tube would change color, therefore allowing the mechanism (the connector tube) to be even more visible.

Because other bird species have succeeded at these tests regardless of whether they are a caching specialist or a tool user, we predict that scrub-jays will perform similarly. Specifically, we predict that they will prefer water-filled rather than sand-filled tubes (as in corvids), and heavy rather than light objects and that they will change their preferences when the heavy objects become non-functional (as in grackles). We also predict that they will not learn to associate color with a reward in the Colored U-tube experiment or attend to the exposed mechanism in the Uncovered U-tube experiment (similar to most corvids). If scrub-jays attend to the functional properties of objects and substrates and flexibly change their preferences when the task changes, this would indicate that their highly discriminatory and flexible behavior generalizes to conditions outside of the context in which their cognitive abilities evolved.

Methods

Animal ethics

This research was carried out in accordance with the University of California, Los Angeles’ (UCLA) Institutional Animal Care and Use Committee (protocol number 1995-026-63).

Subjects

Three wild adult male Western scrub-jays (Aphelocoma californica) were caught using Potter traps baited with peanuts in southern California (July–August 2013), and one female nestling (BB; an adult at the time of these experiments) was taken from the nest in the summer of 2012 and hand-raised (all captures were authorized under appropriate federal and state collecting permits). Birds were sexed genetically (following Griffiths et al., 1998), and the validity of this measure was confirmed via inspection of the gonads by Rensel and colleagues (2015). Moreover, in Rensel et al. (2015), all four scrub-jays successfully participated in caching experiments, thus indicating their acclimation to captivity. Before and after our experiments, scrub-jays were housed socially with 2–4 birds per aviary (except for BB who was housed singly during a period in which she behaved aggressively toward conspecifics). For the duration of our experiments, scrub-jays were housed singly, in visual but not auditory isolation of other birds, in testing aviaries measuring 5.3 × 1.2 × 1.9 m. Scrub-jays had ad libitum access to water and access to food (Roudybush Daily Maintenance Diet, fruit, and mealworms) for a minimum of 15 h every day. Non-testing food was removed before and during testing when testing occurred. Birds were tested in two batches: BB and GG were tested from August 2014 to January 2015 (requiring a total of 1–3 days per bird per experiment, spread out over the course of between 1 and 7 days), and PA and H from June to November 2015.

Experimental set up

After placing birds in testing aviaries, they were first habituated to testing apparatuses and stones. This was accomplished by feeding birds off of the apparatuses in which the relevant openings were covered by tape so the birds could not discover how the apparatus worked. Once birds readily approached and ate food off of an apparatus or stone, they were considered habituated and the experiment began. If a bird needed to be re-habituated in the middle of an experiment, the experiment was paused for this habituation to take place. For each experiment, testing apparatuses were placed on a paper-covered table (0.3 × 1.1 × 0.6 m) inside the aviary with perches placed above the table to allow easier access to the apparatuses. Testing lasted up to five hours per day between 0700 and 1600. If testing occurred in the morning, food was removed from the aviaries the night before (between 1800 and dusk). For afternoon test sessions, food was removed at 0700. Testing sessions lasted up to approximately 20 min. Trials ended if the bird obtained the reward or did not interact with the task after 4 min, at which point the apparatus was removed from the testing aviary for at least 10 min before resuming the session. If a bird did not interact with the task after 2 min, bait (a small peanut piece) was placed on the table equidistant between the tubes (if a two-tube experiment) or objects to encourage the bird to participate. If the previous session ended with no participation in the task, the first trial of the next session began with bait, thus a trial was baited up to two times. An experiment was ended before completion due to a lack of the bird’s motivation if the bird did not interact with the apparatus for five consecutive sessions over the course of multiple days. Water tubes were baited with peanut pieces attached to cork using a tie wrap to allow the food to float (hereafter referred to as peanut floats). Experiments 1–5 consisted of 20 trials per bird per experiment. All experiments were recorded with a Sony Handycam HD camera on a tripod.

Color learning for side bias prevention

To prevent side bias during the water tube experiments involving two tubes, scrub-jays were required to learn to associate food with color, forcing them to attend to color rather than location (as in Logan et al., 2014). A gold tube always contained food (small peanut pieces), while a silver tube never did. One gold and one silver tube were placed on the table, one on the left and one on the right (left side first, pseudorandomized for side) with the open ends of the tubes facing the side walls such that birds could not see which tube contained the food. Birds were habituated to the task using a blue tube (all tubes measured 50 × 50 × 67 mm, outer diameter = 26 mm, inner diameter=19 mm) until they learned to search for food even if it was not visible. After habituation, the color learning test began and scrub-jays got one choice per trial, marked as the first tube they look into, and proficiency was reached when an individual chose the gold tube at least 17 out of the most recent 20 trials (having achieved at least 8 out of 10 on each set of 10 contributing to the passing score). Pseudorandomization consisted of alternating sides for the first two trials and then allowing each tube to remain on the same side for a maximum of two consecutive trials.

Between 20 and 80 trials were required for three birds to reach proficiency, similar to grackles (CJ Logan, 2014–2015, unpublished data), Darwin’s finches (Tebbich, Sterelny & Teschke, 2010), and pigeons (Lissek, Diekamp & Gunturkun, 2002), and faster than pinyon jays, Clark’s nutcrackers, and previously tested Western scrub-jays (Bond, Kamil & Balda, 2007). GG did not complete this training due to his lack of willingness to come to the table to participate in the task. During two-tube water tube experiments (Experiments 1, 4, and 5), a side bias was considered to have developed if a bird approached the same side three or more times consecutively. At this point, the experiment was suspended and the subject was given the color test. If they chose gold at least 8 out of 10 trials, the experiment resumed. However, if they no longer had a color preference, they were tested until they chose gold at least 17 out of the most recent 20 trials (per the criterion above), and then the experiment resumed. Color trials were given to BB after trial 18 in the Colored U-tube experiment and after trial 14 in the Uncovered U-tube experiment. GG did not develop side biases.

Table 2 All choices per trial per bird per experiment.

The order in which the functional and relatively more functional choices (dark gray: water, heavy, light, rewarded color, connected), or non-functional and relatively less functional choices (light gray: sand, light, heavy, unrewarded color, unconnected) were chosen (columns) and whether the bird successfully obtained the food (marked with an X) for trials 1–20 (rows).

Experiment 1. Water vs. Sand	Experiment 2. Heavy vs. Light	Experiment 3. Heavy vs. Light Magic	Experiment 4. Colored U-tube	Experiment 5. Uncovered U-tube	
GG	GG	BB	BB	BB	BB	
	Insertion order		Insertion order		Insertion order		Insertion order		Insertion order		Insertion order	
Trial	1	2	3	4	Trial	1	2	3	4	5	Trial	1	2	3	4	5	Trial	1	2	3	4	5	6	Trial	1	2	3	4	Trial	1	2	3	4	
1					1						1				X		1							1	X				1		X			
2		X			2			X			2				X		2		X					2					2			X		
3			X		3			X			3		X				3		X					3				X	3		X			
4	X				4					X	4				X		4							4			X		4			X		
5	X				5			X			5			X			5	X						5	X				5		X			
6	X				6						6		X				6							6				X	6	X				
7	X				7						7			X			7							7				X	7				X	
8			X		8						8					X	8							8		X			8				X	
9					9		X				9				X		9							9	X				9			X		
10					10						10				X		10							10				X	10	X				
11					11						11	X					11		X					11	X				11	X				
12	X				12	X					12			X			12							12			X		12			X		
13					13	X					13					X	13							13			X		13	X				
14					14						14					X	14							14		X			14				X	
15		X			15	X					15				X		15			X				15					15					
16					16	X					16			X			16	X						16				X	16			X		
17		X			17		X				17	X					17	X						17		X			17					
18		X			18						18	X					18			X				18	X				18			X		
19			X		19	X					19			X			19							19		X			19	X				
20				X	20			X			20					X	20			X				20		X			20					

Stone dropping training

Birds were trained to lift stones off of the testing table, carry them to the perch, and drop them down the tube of an apparatus with a collapsible platform. The apparatus was a clear cast acrylic box (185 × 110 × 85 mm) with a 90 mm tube (outer diameter: 51 mm, inner diameter: 43 mm) on top of the box and a platform inside that was held up by a magnet (Fig. 1A; as in Bird & Emery, 2009b). Magnetic contact was broken upon impact from the stone dropped into the tube, allowing the platform to fall down and release food onto the table. Birds were first encouraged to accidentally push the stone into the tube by placing a small piece of peanut under the stone balanced on the edge of the rim. They then progressed to picking up and dropping the stone into the tube from anywhere on the table. Birds accessed the top of the tube by standing on a perch placed near the top of the tube rather than by standing on the ground because they were more willing to participate in this context. The configuration (e.g., standing on a perch or the ground) should not have influenced the bird’s perception of the task because, in both scenarios, their head was always over the tube when the object was dropped in and birds could move their heads to look at the outside of the tube. Proficient stone drops were defined as those in which the bird picked up the stone from the table and directly dropped it into the tube. Once proficiency was reached, 30 more trials were conducted to ensure their expertise on the task. BB and GG required 76 and 255 trials to pass this training, respectively, while PA and H never passed (we stopped their training at 536 and 507 trials, respectively; Table 2).

Figure 1 Single stone dropping apparatus (A) and multi-stone dropping apparatus (B).

Photo credit: Brigit Harvey.

Multi-stone dropping training

After reaching proficiency on stone dropping training, birds received multi-stone dropping training to learn that solving a task might require dropping more than one object into the tube. The multi-stone apparatus was similar to the stone dropping training apparatus, but had a larger box (box: 200 × 180 × 150 mm; tube: 95 mm tall, 50 mm outer diameter, 44 mm inner diameter; Fig. 1B; as in Logan et al., 2014) and the platform was balanced on a circular rod rather than being held up by a magnet. Counterweights placed at the rear of the platform ensured that 2–4 stones needed to drop down the tube and then slide down a ramp to land on the front of the platform before the platform would fall open, thus releasing the food. Individuals passed this training once they successfully solved 10 consecutive trials. BB and GG were immediately proficient, thus they completed all 10 trials proficiently.

Reachable distance

After multi-stone dropping training, the height at which a bird could reach the food in the tube was determined in advance to establish how high to set the water level in the experiments. This was necessary so that, during experiments, the food would be out of reach and require the desired number of objects to bring it within reach. The reachable distance was the distance from the bottom of the tube to the top of the food, which sat on top of a plastic sandwich bag stuffed with cotton in a standard tube used in the water experiments (a clear cast acrylic tube measuring 170 mm tall, 50 mm outer diameter, 43 mm inner diameter and attached using super glue to a clear cast acrylic base measuring 300 × 300 × 3 mm). Birds were allowed to access the food (peanut floats), initially well within reach, and then the distance was decreased until it was out of reach.

Experiment 1: Water vs. Sand

This experiment consisted of two standard tubes: one partially filled with sand and the other with water to the same height in each tube, to determine whether birds preferred to drop stones into the functional water tube rather than the non-functional sand tube (Fig. 2; similar to Logan et al., 2014). First, birds were given a 10-trial training period in which any initial tube preferences were discouraged by heavily baiting the non-preferred tube. Tubes contained water and sand (and were pseudorandomized for side), but no floating food. The tops of the tubes were taped over and bait (peanut pieces) was placed on top and at the base of each tube. The tube the bird ate from first was recorded to determine whether a preference emerged. After these training trials, the experiment began and the sand and water tubes continued to be pseudorandomized for side. Four stones (weighing 14–21 g and displacing 6–8 mm water each) were located between the two apparatuses: two on the base of one apparatus and two on the base of the other apparatus, and each bird experienced 20 trials.

Figure 2 Water vs. Sand experiment.

Birds were given stones and could choose to drop them into the water-filled (functional) or sand-filled (non-functional) tube. Photo credit: Brigit Harvey.

Experiment 2: Heavy vs. Light

One standard water tube was given with 8 objects: four heavy (a steel rod encased in fimo clay, each weighing 10 g and displacing 2–3 mm of water) and four light (a black plastic tube partially filled with fimo clay, each weighing 2 g and displacing 1–1.5 mm water; Fig. 3A; as in Logan, 2015a). Heavy and light objects were 21–24 mm in length and 8 mm in diameter. Both objects sank, making them both functional, however heavy objects had a larger volume (1,056–1,207 mm3) than light objects (approximately 500 mm3). Therefore, heavy objects displaced more water and were more functional. Volume differences were created by making one end of the inside of the tube hollow for the light objects. First, birds were given a 3-trial object training period without a water tube in which any initial object preferences were discouraged by placing relatively more bait (peanut pieces) on the non-preferred object. A heavy and a light object were placed next to each other on the table (pseudorandomized for location) and bait was placed underneath and on top of both objects. The object the bird ate from first was recorded to determine whether a preference emerged, and the trial ended when the bird had interacted with both objects. After these object training trials, the 20-trial experiment began and pairs of heavy and light objects were pseudorandomized for location.

Figure 3 Heavy vs. Light (A) and Heavy vs. Light Magic (B) experiments.

Birds could drop heavy (more functional) and light (less functional) objects into the water tube (A). They were then given a followup experiment where the heavy objects became non-functional because they stuck to a magnet placed on the tube above the water (notice the heavy object stuck to the magnet), thus making the light objects the only functional option (B). Photo credit: Brigit Harvey.

Experiment 3: Heavy vs. Light Magic

This experiment was the same as Experiment 2, except here the heavy objects became non-functional to determine whether birds could discriminate between the functional properties of the objects and change their preference from the previous experiment. A magnet was attached to the inside of the tube above the water level so that the heavy (metal) objects became non-functional (they stuck to the magnet if inserted into the tube), thus making the light (non-metal) objects the only functional option because they could fall past the magnet and into the water (Fig. 3B). Three heavy and three light objects were placed in pseudorandomized pairs at the base of the tube because four heavy objects would not fit on the magnet. Each bird was given 20 trials.

Experiment 4: Colored U-tube

This experiment consisted of two apparatuses made of clear cast acrylic, each containing a standard tube and a small-diameter tube (small tube outer diameter = 25.4 mm, inner diameter = 19 mm) 25 mm apart, with 160 mm of tube above and 90 mm below a clear cast acrylic lid (300 × 400 × 3 mm) on a wooden box (Fig. 4A). The small tubes contained out of reach peanut floats (the reachable distance for each bird was obtained for the small tube prior to beginning the experiment), but were too small for stone insertion. On one apparatus, a tube under the lid connected the two water tubes such that inserting the stone into the standard tube resulted in the food rising in the small tube. The connected apparatus was indicated by a particular color (counterbalanced across birds) and was pseudorandomized for side. The apparatuses were the same as in Logan et al. (2014) with modifications to make the two tubes on each apparatus appear as part of the same apparatus and to distinguish the two apparatuses from each other. Instead of both apparatuses having a white paper background with differently colored shapes at the base of the standard tube, here each apparatus had a distinct background color (blue or brown). On top of these backgrounds, each apparatus had a different color and shape (pink triangle or yellow square) that extended around the base of the two tubes to further unify the tubes of each apparatus by making them appear more as a single unit, instead of only extending around the base of the standard tube. The tops of each apparatus’ tubes were marked with electrical tape identical to the colored shape at the base (pink or yellow). One white strip of electrical tape was placed on each apparatus to indicate that these are the same apparatuses in the next experiment. Any initial apparatus preferences were discouraged by heavily baiting the non-preferred apparatus over the course of 10 trials as in Experiment 1. Four stones were placed between the apparatuses as in Experiment 1 and 20 trials were given to each bird.

Figure 4 Colored U-tube (A) and Uncovered U-tube (B) experiments.

Birds were given stones that they could drop into the tube of the color that indicated the connected (functional) apparatus or the unconnected (non-functional) apparatus (A). In a followup experiment, the connector tube was visible and birds could choose to drop stones into the connected (functional) or unconnected (non-functional) apparatuses (B). The connector tube is visible on the apparatus on the right in (B). Photo credit: Brigit Harvey.

Experiment 5: Uncovered U-tube

This experiment was the same as Experiment 4 except all paper and color cues were removed and the boxes around the bases were removed from both apparatuses, thus exposing the connector tube under the lid of the connected apparatus (Fig. 4B). During the experiment, 20–30 drops of red (for pink) or yellow food coloring (the same as the colored paper and tape on the connected apparatus in Experiment 4) were placed into each wide tube such that when a stone was dropped into the connected apparatus, the flow of tint from the standard to the small tube would show the water flow through the connector tube. Note that the unconnected apparatus had alternate dye patterns during BBs first 3 trials before settling on this methodology: there was red dye in trials 1 and 2, and no dye in trial 3.

Statistical analyses

Binomial tests were carried out in R v3.2.1 (R Core Team, 2015), and, when there were multiple p-values per experiment, they were corrected using the Bonferroni–Holm method.

Results

All choices per trial per bird are shown in Table 2 and a video showing examples of the experiments is available online at: https://youtu.be/KmLnVqPDrZ8. During the pre-experiment trials to control for preferences, there was no preference across birds in their first trial for one object/tube or the other (see data at KNB; Logan, 2015b).

Experiment 1: Water vs. Sand

GG had no preference for which tube to drop the stones into (Table 3). BB did not complete this experiment. Her motivation to participate declined, possibly because she received few food rewards (she primarily chose the sand tube). To prevent her from giving up on dropping objects down tubes entirely, she was given alternating sessions with either a single water tube or the multi-stone dropping apparatus and stones for four days until she began regularly interacting with the water tube again.

Table 3 Summary of results.

The number of trials required to learn to associate food with the gold tube (color test; min. 17 out of 20 trials correct) and to become proficient at dropping stones down the platform apparatus (stone drop training; number of non-proficient stone falls plus 30 proficient stone drops); total number of correct choices/total number of choices and p-values from binomial tests for experiments 1–5 (the Bonferroni–Holm correction was applied to Experiment 2).

Bird ID	Sex	Color test	Stone drop training	Exp 1: Water vs. Sand	Exp 2: Heavy vs. Light	Exp 3: Heavy vs. Light Magic	Exp 4: Colored U-tube	Exp 5: Uncovered U-tube	
BB	F	80	76	X	33/56 0.46	42/68 0.06	30/52 0.33	32/53 0.17	
GG	M	X (28)	255	29/51 0.40	27/45 0.46	–	–	–	
H	M	20	X (507)	–	–	–	–	–	
PA	M	50	X (536)	–	–	–	–	–	
Notes.

X bird did not complete this experiment

– bird did not participate in this experiment

Experiment 2: Heavy vs. Light

BB and GG consistently successfully obtained the food using both heavy and light objects without a preference for the more functional heavy objects (Table 3).

Experiment 3: Heavy vs. Light Magic

BB had no preference for heavy or light objects, though it appeared that she was developing a preference for light objects near the end of her experiment so it is possible that the preference would have been significant given more trials (Table 3). GG stopped participating in experiments at this time at first because of his neophobic reaction to the magnet, but even after a successful magnet habituation period, his motivation for participating in tests did not recover.

Experiment 4: Colored U-tube

BB had no preference for dropping stones into the standard tube on the brown apparatus, which indicated the connected apparatus (Table 3).

Experiment 5: Uncovered U-tube

BB had no preference for dropping stones into the connected apparatus (Table 3).

Discussion

Two scrub-jays successfully obtained the food in the Heavy vs. Light experiment because both objects were functional, however, contrary to predictions, no scrub-jays attended to the functional differences between objects or tubes or changed their preference when the task changed. In every other species tested so far, including a caching specialist (Eurasian jay), at least some individuals attended to the functional differences between objects and/or substrates, thus making the scrub-jays the first species tested to fail to demonstrate such attention to function (Bird & Emery, 2009a; Cheke, Bird & Clayton, 2011; Taylor et al., 2011; Jelbert et al., 2014; Logan et al., 2014). While it appeared that BB was learning to prefer light objects in Heavy vs. Light Magic, she did not learn to significantly prefer this object type within the 20 trials that are standard for these experiments. Consistent with predictions, scrub-jays performed similarly to most previously tested corvids and failed the U-tube tests. Failure on the Colored U-tube task could indicate that a degree of causal cognition is used to solve this problem because attention to causal cues and expectations about causal relationships could inhibit learning to associate arbitrary cues with a food reward (Cheke, Bird & Clayton, 2011; Jelbert et al., 2014; Logan et al., 2014). The color and water flow modifications to the Colored U-tube and Uncovered U-tube experiments did not appear to facilitate learning to prefer the connected apparatus.

Our modifications to the Heavy vs. Light experiment likely made it more difficult to solve. In all species previously tested, except grackles, the heavy objects sank and the light objects floated, therefore functionality was dichotomous (Bird & Emery, 2009a; Cheke, Bird & Clayton, 2011; Taylor et al., 2011; Jelbert et al., 2014; Logan et al., 2014; Logan, 2015a). In the grackle (Logan, 2015a) and scrub-jay experiments, both items were functional, but heavy objects were approximately twice as effective as light objects. This modification allowed a followup experiment (Heavy vs. Light Magic) within the Aesop’s Fable paradigm to test behavioral flexibility. However, this modification had other consequences. First, it made functional discriminations between heavy and light objects more difficult: there was a smaller difference in the functionality of the objects because both objects sank, thus birds were not forced to discriminate between objects to obtain the food. The followup experiment, Heavy vs. Light Magic, was designed to test attention to the functionality of the objects, and in this case, the one scrub-jay who participated in both of these experiments exhibited no preference in either test. Second, the functionality of the light object was disassociated from its weight because the smaller volume is what caused it to displace less water. This means that birds could solve the task by associating light objects with receiving food due volume differences or by using the methods proposed for other species if they bind the association of volume with weight. This potentially made the task more difficult. Additionally, the Heavy vs. Light Magic experiment was more difficult than other experiments because heavy items that were dropped into the tube stuck to the magnet and blocked access to the floating food reward. Therefore, a bird should inhibit dropping any heavy objects and switch to only dropping light objects into the tube to more easily obtain the food, thus making this task particularly difficult. Perhaps scrub-jays would have passed the easier version (sinking vs. floating) of this experiment if they were given the opportunity.

That both objects were functional in the Heavy vs. Light experiment allowed us to begin to examine an alternative explanation for how individuals solve Aesop’s Fable tests: the object-bias hypothesis (Logan et al., 2014; Jelbert, Taylor & Gray, 2015). In previous Heavy vs. Light experiments where heavy items sank and light items floated, all individuals preferred the heavy items. This could indicate that they attended to the functional differences of the objects or that they had an innate bias toward the heavy objects perhaps because they were more similar to objects in the wild such as stones. Since both scrub-jays tested did not have an object preference in the modified design, this suggests that object biases might not drive their choices. That scrub-jays can learn to drop stones down tubes and successfully obtain a food reward in the Heavy vs. Light experiment is further confirmation that non-tool using species are capable of this tool use task (Table 1). The two scrub-jays that became proficient at stone dropping required a similar amount of training as required by grackles (6 grackles learned in 135–362 trials and 2 grackles never learned; Logan, 2015a) and New Caledonian crows (6 crows learned in 1–116 trials and others never became proficient, CJ Logan, 2013, unpublished data). Using the same platform apparatus, Eurasian jays and rooks needed far less training to learn the task (4 Eurasian jays learned in 11–33 trials and 1 never became proficient, Cheke, Bird & Clayton, 2011; whereas all 4 rooks learned in 5 trials, Bird & Emery, 2009b). Two out of 4 scrub-jays never became proficient at stone dropping, and thus did not participate in stone dropping experiments, and both scrub-jays that participated in experiments did not participate in every experiment. It appeared that their lack of motivation for participating in these kinds of tasks slowed their learning and could have caused them to give up; alternatively they could have lacked motivation due to cognitive limitations preventing them from solving the tasks. The exception was BB who showed more motivation than the others and participated in more experiments, perhaps due to her being the only hand-raised scrub-jay in this group—a developmental experience that has been shown to improve cognitive performance in other species (see Thornton & Lukas, 2012).

Limitations and future directions

The scrub-jays’ lack of motivation combined with their lack of a preference for the functional options suggests that either the Aesop’s Fable paradigm is too ecologically irrelevant for this species or that their highly discriminatory and flexible behaviors do not transfer to a non-caching context. Alternatively, it is possible that this species is capable of such discriminations: perhaps the individuals in our small sample were not discriminatory but others might be. While other studies using the Aesop’s Fable paradigm also had small sample sizes, at least two individuals from each study made some functional discriminations. For example, in the Heavy vs. Light experiment, four grackles preferred heavy objects and two had no preference—the latter two grackles performing similarly to the two scrub-jays. That we were only able to test two scrub-jays (and usually only 1 per experiment) opens the possibility that we did not capture the range of individual variation possible for this species in these experiments. Future studies using different non-caching paradigms are needed to determine whether scrub-jays’ cognitive abilities transfer to non-caching contexts.

We thank Rick Klufas in the UCLA Bioshop; Joe Jablonski at the UCSB workshop; Russ Revlin for his idea to tint the water in the U-tube experiment; Alison Greggor for her suggestion to use the tint to indicate water flow; Rachael Shaw for her idea to use white tape as an indicator of U-tube apparatus continuity; and Sarah Jelbert, Zoe Johnson-Ulrich, Jennifer Vonk, and an anonymous reviewer for helpful feedback on a previous draft of this manuscript.

Additional Information and Declarations

Competing Interests

Author Contributions

Animal Ethics

Data Availability

The authors declare there are no competing interests.

Corina J. Logan conceived and designed the experiments, analyzed the data, contributed reagents/materials/analysis tools, wrote the paper, prepared figures and/or tables, reviewed drafts of the paper.

Brigit D. Harvey performed the experiments, prepared figures and/or tables, reviewed drafts of the paper.

Barney A. Schlinger contributed reagents/materials/analysis tools, reviewed drafts of the paper.

Michelle Rensel reviewed drafts of the paper, provided logistical support.

The following information was supplied relating to ethical approvals (i.e., approving body and any reference numbers):

This research was carried out in accordance with the University of California Los Angeles’ (UCLA) Institutional Animal Care and Use Committee (protocol number 1995-026-63).

The following information was supplied regarding data availability:

Logan CJ. 2015b. Western scrub-jay water tube experiments, Los Angeles, CA USA 2014–2015. The Knowledge Network for Biocomplexity (KNB). https://knb.ecoinformatics.org/#view/doi:10.5063/F10C4SP7.

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
