# Peer review of "Western scrub-jays do not appear to attend to functionality in Aesop’s Fable experiments"

_PeerJ, doi:10.7717/peerj.1707_

## Round 0.1 · original submission · Major Revisions

The MS was reviewed by three reviewers with unique expertise and each reviewer has found merit in the studies and in your presentation of them. However, each of the reviewers have raised a number of issues that should be addressed before the paper can be accepted for publication in PeerJ. I agree with Reviewer 3 that further experimentation is not necessarily needed, but the points raised by this reviewer, and the others, should be addressed in a limitations and future directions section. Please take care to add the information requested by the reviewers and be clear in your rationale and predictions for the study in the revision.

Reviewer 1 ·

Basic reporting

I feel this is a relatively clear, concise introduction to topic and study species, however I do have some suggestions for improvements.

I would split various paragraphs/ sentences into smaller paragraphs/ sentences. E.g. Split paragraph 1 into 2 paragraphs, at line 35, paragraph 2 into 2 smaller paragraphs, at line 61, paragraph starting line 113 split at line 126. E.g. Split sentences e.g. lines 67-71 into 2 sentences, and lines 77-80 into 2 sentences, lines 143-146.

Lines 41-46. Reference to 3-4 of 4 rooks, 0-2 of 5 EJs, 1 of 4 water vs sand and 4 of 4 other experiments on group averages NCC, 0-6 of 6 NCC. I find this the use of 0 – number confusing and just listing “X experiments” isn’t clear one what these actual experiments involved and how it relates directly to the current study. It’s also not clear just from this info if any of the scrub-jay experiments were new, or rather replications of previous studies. I’d suggest you include a table with these prior results i.e. number of subjects per species solving each experiment, and the type of experiments used in each previous study (e.g. sand vs water), as opposed to listing this information within the text. This should help in making it clearer why you chose these particular experiments for the scrub-jays. You could also refer to it when you discuss your small sample size in the discussion - this is one of my suggestions later on.

I would also emphasize when referring to other corvids/ species tested with AF task whether they are tool users in the wild or not, since this is a test that requires tool use and not all readers will be familiar with corvid capabilities in this context. Again, this should help for the discussion, as I think this point (tool users vs non-tool users in a tool use task) needs to be discussed. You could even include this in the table suggested above alongside caching specialist/ not to emphasize this latter point as well.

Experimental design

I feel you need to further justify why you chose this particular paradigm to test non-caching context cognition in scrub-jays - it's a tool use task and this is a non-tool using species. If you're only interested in behavioural flexibilty, there are other simpler paradigms you could have used with this species. This needs further justification in intro and discussion later on.

Line 77-80 – I would appreciate clearer predictions for each experiment, particularly for experiment 4 and 5. This could be incorporated in a figure of the apparatus e.g. “Correct/ functional choice is…”, after each experiment description or a separate table.

Line 109 – 20 trials per bird per experiment?

Line 112 – I like that you included this control for development of side biases.

Line 113 – in line 28 you state you will refer to the birds as “scrub-jays” not “jays” as you do in line 113, please make sure you do this through-out the ms for consistency.

Line 130 – “lack of motivation” as determined by what? Did you include any sort of motivation test?

Lines 138 – no reference to habituation to apparatus, did you include this? You refer to an issue of a subject being afraid of the magnet (lines 269), could lack of habituation be a general problem? I don’t think so necessarily, but please clarify when and how you habituated the birds to the apparatus/ objects – or if you needed to do this.

Lines 146 – The birds perched on top of the tube to drop the stone, rather than the ground (I believe the other species solved from the ground). One of the critiques of the AF task is the possible role of feedback (stone drops into water, water level and therefore reward incrementally rises closer to subject). If the subject stands above as opposed to on the side of the tube, could this be a potential explanation for the failure of the task – assuming the birds do rely on feedback? I’m not sure, but this point could be worth discussing.

Line 188 – Add “on the other apparatus base”.

Line 297- “far less training” rather than “much”

Validity of the findings

Lines 255 – in the pre-test trials where you heavily rewarded the non-preferred object/ apparatus, did the birds show any preferences from trial 1, i.e. first time they interacted with each set of tubes/ objects?

Lines 302 – as mentioned earlier, how did you determine that the issue was lack of motivation? Were they sufficiently habituated to the apparatus/ objects prior to testing? Were they comfortable enough in captivity (as wild birds) to participate in these experiments?

Lines 306 – and was she a juvenile too? Could this have influenced your results?

In the discussion, there are various other potential limitations to this study that you need to discuss. For instance, the light objects being functional if enough were inserted may influence any preference for heavy over light. The exceedingly small sample size - only 2 subjects in experiment 1-3 and only 1 subject in experiment 4 and 5 – must at least be acknowledged. Did you test for motivation and successful habitation to the apparatus/ objects prior/ during experiments? Could dropping the stone from the top of the tube rather than the ground have influenced results (see comment above)? Could giving experience of taking a bait from the top of a closed tube (e.g. experiment 1) have reduced motivation once the tube was open? Use of a tool use task in a non-tool using species – this needs to be discussed with reference to other species and previous results.

·

Basic reporting

The manuscript is well written. To put the results in context, the introduction requires more detail regarding the Aesop’s Fable tasks and what they have been used for in previous studies. In particular, Aesop’s Fable’s tasks have typically been used to investigate causal understanding, rather than behavioural flexibility. In previous studies different tasks were presented that either tested the birds understanding of different aspects of water displacement, or, alternatively, provided causal, arbitrary or counter-intuitive cues to determine whether the birds used different types of information equally (e.g. Cheke et al). Therefore, in previous studies there are some experiments which subjects are expected to pass and others that they are expected to fail. This is currently overlooked in the introduction by stating that ‘some to all individuals tested using the Aesop’s Fable paradigm were successful…’ and providing only summaries of the number of tests given, and the range of subjects that passed on any test, rather than the purpose of these tests (lines 38-49). I would recommend describing the tasks most similar to the ones used in the current study, and stating how many subjects passed these particular tasks. This information is already summarised in a table in a mini-review I wrote, which could be useful here (Jelbert et al. 2015 Comm Int Bio). See also the comments in ‘validity’ below.

Related to this, it is not clear what the specific predictions are for this study. While I agree that success on the paired tasks would indicate behavioural flexibility, based on the performance of other corvids it would be reasonable to expect that scrub jays might pass sand vs. water, but not either version of the U-tube.

Experimental design

The methods are generally detailed and well described. There are a few areas where more information is needed:
- The number of trials given is stated at the start of the methods but it would be helpful to remind the reader when describing each task (e.g. lines 186-188).
- What were the criteria for ending a trial (obtaining the reward / time limit / leaving the table)?
- In the video for sand vs water the tubes were baited a second time when the bird did not return to the table. What were the criteria for this? Were other methods used to increase the birds’ motivation?
- What were the criteria for ending an experiment (in the case of low motivation)?
- How long did each experiment take to run per bird, were the trials spread out over days/weeks?

Lines 233-236: It is stated that the coloured U-tube contained food colouring, however the water in the video and in the figure is not coloured. Was the water coloured for some trials but not all? If so, why?

Validity of the findings

The findings from this paper are based on the performance of only 1-2 birds on each test. Given the very small sample size, and the fact that all results were negative, the conclusions need to be stated very cautiously. I think the authors generally do a good job with this, but they need to consider other explanations for the scrub jays poor performance. In particular, apart from the sand vs water task (which followed the same procedure as other studies), the experiments provided here could have been particularly difficult to pass. Variations of the U-tube task have been used in three studies, with almost unanimous failures (0/2 Eurasian jays, 0/4 NC crows, then 1/5 NC crows passed). Thus, the scrub jays performance is entirely in line with the performance of other corvids, who also fail.

The light vs heavy task (and light vs. heavy magic) has only been previously used with grackles. The other corvid species have been tested with a sinking vs. floating task, which is superficially similar to the current study (because both involve heavy and light objects), but is functionally dissimilar because in the current study both objects sink. Here, functionality is unrelated to weight: one object displaces less water presumably because it is partially hollow (it is unclear from the ms exactly why the light object displaces less water), but this is not a general feature of light or heavy objects. The lack of a clear causal relationship between weight and functionality may have made the task harder to pass. Furthermore, the incentive to choose only heavy objects is lower in this version, where both objects are functional, than when light objects don’t have an effect on the reward (e.g. for NC crows provided with polystyrene that floats above the water). These task differences need to be addressed when drawing comparisons between the performance of the scrub jays and other species.

Given that the results are easy to summarise I would also recommend stating them explicitly in the abstract (i.e. along the lines of: 4 birds were tested but only 2 learnt to drop stones. Of these one bird took part in 4/5 experiments and one in 2/5 experiments. Neither bird passed any task).

Other points:

Lines 247-248: I don’t think the use of Bonferroni-Holm corrections is appropriate here. It raises the passing criterion for one experiment only: heavy vs light, where both birds took part. It would be fairer to examine which tests each bird can (hypothetically) pass and which ones they fail, by comparing each bird’s performance, on each task, to a consistent uncorrected P-value.

Table 1: Please provide the bird’s choices as well as p-values

Supplementary Figure: This is a very clear way of displaying the main results. I would recommend including this in the main article rather than the supplementary material.

Colour learning for side bias prevention: How often was this applied in practice? Could a side bias have contributed to the bird’s poor performance?

Additional comments

Lines 255-257 & 268-270: This is speculative and should be reworded.
Lines 295-296: The number of trials to learn stone dropping were not recorded in Jelbert et al 2014.
Figure S1: legend describes more or ‘less’ functional choices, but should be ‘non-functional’ for most tasks.

·

Basic reporting

The article is clear, well written, and self-contained; it uses appropriate structure and figures.It includes sufficient background information to demonstrate how the work fits into the broader field of knowledge: the facts that scrub jays show multiple cognitive adaptations within a caching context (flexibility, planning, cache protection strategies) that could conceivably be applied outside this context, and that many other corvid species (and some non-corvids) as well as other caching species have passed some elements of the Aesop's Fable paradigm creates a strong hypothesis that scrub jays will also be able to pass this test. In addition it supports the idea that whether or not scrub jay's cognitive abilities are generalizable is of interest, given that their cognitive adaptations for caching are impressive.

Experimental design

The submission describes original primary research within the scope of the journal. The submission clearly defines the research question: can western scrub jays generalize their cognitive adaptations for caching to other contexts (in this study, the Aesop's Fable paradigm)? The authors state that scrub jays have not been tested outside of caching contexts, and cite a sampling of studies on scrub-jay caching, as well as other corvid performance on the Aesop's Fable. The investigation was conducted rigorously and to a high technical standard; the inclusion of multiple variations of the water-tube task (as well as the use of other stone-dropping tasks to acclimate scrub jays to stone-dropping) is commendable. The care taken to ensure subjects did not develop side-biases through the use of the colored tubes task is one method that shows the rigor of this study; the authors would not let jays fail simply because of side-biases and took great care to create a testing environment that would allow scrub jays to show their abilities to understand or learn the Aesop's Fable paradigm if they were capable of doing so. The colored u-tube task in particular is a good example of a task that probes causal reasoning versus associative reasoning; while the scrub jays in this study did not perform well enough to require a probe into causal reasoning on the earlier tasks, it is still shows the rigor of the experimental design implemented here.

However there are points where their methods are not completely clear. For example, it isn't immediately clear that the bait used was a peanut with cork on water experiments except for "reachable distance" training (where food was placed on top of a cotton-stuffed plastic sandwich bag). What is the reason for this difference in bait-use? I would assume that it would have helped the subjects to have been exposed to the bait they would see in testing during this first experience with water-tubes, so the author's justification would be appreciated, even if it is something like an ease of measurement issue. Did both birds that started water-tube testing participate in the reachable distance testing?
In addition, it is unclear how birds always had access to food for at least 20 hours every day if food was removed between 1800 and dusk (assuming this is up to 2100 for summer months) for morning testing (which could have lasted 5 hours). This means a bird would be without food from at the latest 2100 to 1200 the next day (which is 15 hours without food except that gained during testing). No matter how the 24 hour cycle of a "day" is construed as starting and ending, there must have been days when birds had much less than 20 hours of access to food. The food restriction per se is not part of this critique, just the description of how long subjects had access to food and water each day. Indeed, because for afternoon sessions of testing, food was removed at 0700, it is implied that birds were food restricted for at least 5 hours before any testing.
Lastly, in the results for Experiment 1, the authors state that BB was given alternating sessions with the stone-dropping apparatus and a water tube until she regained motivation. Was this a single water tube presented alone? How was motivation defined in this context? Performance on the stone-dropping apparatuses, the water tube, or both?

Aside from these points, the methods are easy to understand (particularly with the photographs provided of set-ups) and appear easily reproducible.

Validity of the findings

The findings and discussion are generally strong, but the authors do not discuss the issue of their small sample size influencing both performance and motivation.

Given that only one experiment had multiple subjects participate (and this only 2), it is a little strong to say that scrub jays' flexibility must be restricted to caching contexts or that the Aesop's Fable paradigm is too ecologically irrelevant. The fact that the hand-raised jay showed the highest motivation suggests that being wild-caught, rather than cognitive limitations, could still be the reason for the other subjects' lack of motivation and does not necessarily indicate lack of ability or ecological relevance. It does show lack of ability for wild scrub jays to exploit novel food sources, but not that the abilities aren't generalizable, just that they lack motivation to try and use their abilities in novel contexts.

BB's performance, and the performance of both GG and BB on Experiment 2, is simply not enough to be definitely representative of all scrub jays. The fact that other studies (the one with grackles, for example, found 4 out of 6 birds preferred to drop heavy objects over light objects) often had 1 or 2 birds that did not learn or discriminate on Aesop's Fable tasks reveals the limitations of this sample size. Even 2 or 3 more subjects participated in the tasks (perhaps all hand-raised) would increase the reliability of these findings and the validity of the author's conclusions substantially.

I would suggest repeating the experiment with either a fresh sample of wild-caught jays or hand-raised jays and perhaps eliminating Experiment 3 to avoid the problem that occurred with GG before concluding the ecological irrelevance of the Aesop's Fable paradigm or lack of ability to generalize within scrub jays. I still find the article to be publishable within the scope of PeerJ, but the authors should temper their conclusions to note that it is also possible that with more subject participation, scrub jays may show different behaviors on these tasks. Including the speculation that the current performance is due to ecological irrelevance or an inability to generalize to non-caching contexts is still suitable, but shouldn't be presented as the only explanations.

---

## Round 0.2 · Minor Revisions

As an AE, I have greatly benefitted from three very expert and thorough reviews on the previous version of your MS. I agree that the reviewers have greatly assisted in getting this MS into a publishable form. Your responsiveness to their comments and suggestions is very much appreciated. However, I have a few remaining points of clarification that I would like to see addressed before I can issue a formal acceptance of the MS.

On lines, 55-56, you write that an individual using causal cues to solve tube experiments should fail to solve counter-intuitive experiments where cues are non-causal. This statement requires more care; it is possible to learn to solve a task involving irrelevant or non-causal cues, provided the cues are predictive and can be learned via associative mechanisms. If you mean they do not succeed on the first trial, please clarify. Are the non-functional cues really “counter-intuitive” or just not intuitively informative?

On line 59, I’m not sure why you wrote “the family including…” rather than listing the specific species tested. This is a bit confusing.
On line 114, please place commas around “to succeed”. Please explain the purpose of the connector tube explicitly.
On lines 124, 257, 477, and 488, 504, please change “Since” to “Because”. Please reserve the use of “since” and “while” for temporal indications.
Please insert a space between “every day” on line 172.
Were the birds housed socially when not participating in experiments (line 178)?
What does it mean to “heavily bait the non-preferred object” (line 329) for Exp. 2 if the birds were choosing objects to drop into a single tube? Please clarify.
On line 458, I would change “lack” to “fail to demonstrate”. It sounds like a nitpicky change but, given the small sample size, I’d emphasize that failure to find evidence here does not indicate that evidence does not exist (as you state appropriately in the limitations section).
I didn’t follow why failure with the U-tube tests indicated a reliance on causal information. Please clarify.

---

## Round 0.3 · accepted · Accept

Thank you for your quick and helpful responses to my previous requests for clarification. I am now happy to accept the MS for publication in PeerJ. I think it will make a nice addition to the literature on causal reasoning in corvids.